# Antibacterial Effects of Bacteriocin PLNC8 against *Helicobacter pylori* and Its Potential Mechanism of Action

**DOI:** 10.3390/foods11091235

**Published:** 2022-04-25

**Authors:** Ying Liang, Jiaqian Yan, Ziqi Chen, Qing Gu, Ping Li

**Affiliations:** 1College of Food Science and Biotechnology, Zhejiang Gongshang University, Hangzhou 310018, China; liangying0221@163.com (Y.L.); yanjiaqian27@163.com (J.Y.); cikkychen@hotmail.com (Z.C.); guqing2002@hotmail.com (Q.G.); 2Key Laboratory for Food Microbial Technology of Zhejiang Province, Hangzhou 310018, China

**Keywords:** PLNC8, *H. pylori* ZJC03, bacteriocin, action mechanism, in vitro

## Abstract

*Helicobacter pylori* (*H. pylori*) is a bacterium that can cause a variety of gastric diseases. Most bacteriocins have gained popularity due to their non-toxic effects on cells and antibacterial effects against a wide range of pathogenic bacteria. In this study, the chemical synthesis of the bipeptide bacteriocin PLNC8 was used to investigate its possible action mechanism against *H. pylori* ZJC03 in vitro. Results showed that PLNC8 had significant anti-*H. pylori* ZJC03 potential, which resulted in a significant reduction in urease activity and a minimum inhibitory concentration (MIC) of 80 μM. PLNC8 inhibited the growth of *H. pylori* ZJC03, disrupting its structure as observed by scanning electron microscopy (SEM) and transmission electron microscopy (TEM). In addition, PLNC8 decreased the ATP level and hydrogen peroxide sensitivity of *H. pylori* ZJC03. In conclusion, PLNC8 disrupts the ability of *H. pylori* ZJC03 to alter the host environment, providing a new avenue for the prevention and control of *H. pylori* infection, providing a theoretical foundation for further elucidation of its regulatory mechanism.

## 1. Introduction

Nowadays, *H. pylori* is one of the most frequent bacteria in the epithelial mucosa of the stomach, and it exists in more than half of all human beings [1]. *H. pylori* is estimated to have coexisted with mankind for at least 100,000 years. Currently, *H. pylori* is considered to be the fundamental driver of chronic gastritis (CG), gastric ulcer (GU), duodenal ulcer (DU), gastric carcinoma, and other stomach-related diseases [2]. The International Agency for Research on Cancer (IARC) listed *H. pylori* as a class I carcinogen in 2017 [3]. Almost all people with *H. pylori* infection have gastritis, but in most cases, this tissue gastritis does not create observable symptoms. Approximately 10–15% of infected people develop a gastric or duodenal ulcer. Even so, gastric cancer is still a relatively rare result of infection, occurring in less than 2% of infected people [4].

Antibiotic therapies based on triple therapy (proton pump inhibitor + two antibiotics) and quadruple therapy (based on triple therapy + metal bismuth) are currently the main treatments for *H. pylori* infection [5]. These two approaches both rely on the therapeutic influence of antibiotics on *H. pylori* to a great extent. Over the last three decades of antibiotic therapy of *H. pylori*, the strain has gradually developed resistance to existing antibiotics [6]. Although *H. pylori* can be eradicated to a certain extent, the eradication rate is only 60–80% [6]. The long-term consumption of antibiotics has side effects such as diarrhea, nausea, and appetite disorder [7]. Therefore, new strategies are urgently required for the prevention and treatment of *H. pylori* [8].

In recent years, there has been growing evidence that probiotics may decrease the effects of antibiotics through their metabolites and their own physiological properties and may be utilized accordingly to protect against gastrointestinal infections [9]. Probiotics are therefore considered as one of the alternatives to antibiotics. Studies have shown that *Lactobacillus* is generally regarded as safe (GRAS), and it has been allowed and widely used in clinical trials for the prevention and treatment of *H. pylori* infection [10,11]. It was encouraging to discover that one of the foremost reasons for the functional role of probiotics in the gastrointestinal tract was their metabolites [12]. In 1925, Gracia A. first reported the production of a Φ bactericidal substance known as colistin by the *E. coli* V strain, and since then, bacteriocins have been in public view and continuously studied [13]. Recently, studies have shown that bacteriocin may inhibit *H. pylori*. For example, peptides extracted from some *Lactobacillus* cultures had obvious inhibitory effects on the growth of *H. pylori* [14]. The peptide purified from short-chain *Lactobacillus* BK11 reduced the urease activity of *H. pylori* and significantly decreased the number of *H. pylori* cells adhered to the cultured human gastric adenocarcinoma cell line monolayer [15]. Kim et al. showed that at least seven kinds of LAB bacteriocin had inhibitory functions on *H. pylori* strains [16]. Pediocin BA28 (500 μg/mL, produced by probiotic *Streptococcus acidophilus* BA28) was shown to have a certain inhibitory effect on *H. pylori* and completely eradicated *H. pylori* within 24 h [17]. The bacteriocin Bulgarian BB18 produced by the *Lactobacillus delbrueckii* subspecies showed an inhibitory region of 11–12 mm [18].

Since antibiotics are becoming less effective, bacteriocins with antimicrobial activity are attractive candidates in human medicine. They have the favorable characteristics of displaying low toxicity toward eukaryotic cells, being considered safe and harmless to humans, and exhibiting activity against pathogenic bacteria that have acquired resistance to antibiotics [19]. PLNC8 is secreted by *Lactiplantibacillus plantarum* [20]. PLNC8 α and β are short peptides, composed of 29 and 34 amino acids, respectively. PLNC8 αβ displays dual effects by acting as a potent antimicrobial agent against *P. gingivalis* and as a stimulatory factor promoting cell proliferation [21]. Khalaf H. et al. also reported similar results [22]. Unfortunately, there have been no comprehensive studies on the mechanism of action.

*L. plantarum* ZJ316 is a probiotic strain with broad-spectrum antibacterial properties and the ability to secrete bacteriocins, and its complete genome sequence has been analyzed [23,24,25,26]. It was found that *L. plantarum* ZJ316 could secrete bacteriocin such as PLNC8. In previous research, PLNC8 was shown to have a certain inhibitory effect on the majority of G^+^ bacteria and some G^-^ bacteria [27]. Therefore, to further explore the antibacterial effect of PLNC8 on *H. pylori* ZJC03 and its mechanism, the inhibition zone and minimum inhibitory concentration (MIC) of PLNC8 on *H. pylori* ZJC03 were detected. In addition, urease activity, relative ATP levels, hydrogen peroxide sensitivity, scanning electron microscopy (SEM) images, and transmission electron microscopy (TEM) images of *H. pylori* ZJC03 treated with PLNC8 were also examined. This study supplied a theoretical foundation for studying bacteriocin PLNC8 as an alternative means of preventing and treating *H. pylori* ZJC03 infection.

## 2. Materials and Methods

### 2.1. Synthesis and Preparation of PLNC8

PLNC8 α and PLNC8 β were prepared by solid-phase synthesis through GL Biochem Ltd. (Shanghai, China). Using Fmoc (9-fluorene methoxycarbonyl) polypeptide solid-phase synthesis method, the amino acid carbonyl was first connected with the resin and synthesized in the order from C-terminal (carbonyl terminal) to N-terminal (amino-terminal). The adjacent polypeptides were connected through peptide bonds, and the resin connected with the complete polypeptide was obtained. Then the synthesized polypeptide was cut off from the resin to obtain the crude polypeptide. The peptide was detected by mass spectrometry. After the qualified product was determined, the target product was separated and purified by RP-HPLC, got purified product PLNC8 α (98%) and PLNC8 β (98%). PLNC8 α and PLNC8 β were dissolved in 0.05% acetic acid buffer respectively, mixed PLNC8 α and PLNC8 β in equal amounts to obtain PLNC8 stock solution, and stored at −80 °C.

### 2.2. Bacterial Strains and Cultural Conditions

*H. pylori* ZJC03 was separated from patients with gastritis and peptic ulcer at the Second Affiliated Hospital Zhejiang University School of Medicine, and cultured on Columbia blood agar (CBA) plates (Hopebio, Qingdao, China) with selective supplement (Hopebio, Qingdao, China) and 7% Sterile Defidrinated Sheep Blood (yuanye Bio-Technology, Shanghai, China) at 37 °C for 72 h with the anaerobic box (85% N_2_, 10% CO_2_, 5% O_2_) (Mitsubishi Gas Chemical, Tokyo, Japan) containing micronutrient bags (Hopebio, Qingdao, China) [28]. *H. pylori* ZJC03 is included in a China Typical Culture Collection strain (CCTCC NO: M 20211218). The strains were classified by 16S rRNA gene sequencing with primers (the forward primer 27F and the reverse primer 1495R) [29].

### 2.3. The Diameter of the Inhibition Zone of PLNC8

The PLNC8 inhibitory zone size was defined by the filter paper (6 mm) diffusion method. 100 μL of *H. pylori* ZJC03 cells (OD_550_ = 0.50 ± 0.05) were spread evenly on the mediums, and filter paper sheets were attached equidistantly to the plates containing *H. pylori* ZJC03 on the surface using sterile forceps [30]. Equal volumes of PLNC8 with different concentrations were dropped on the filter paper accurately with a pipette and incubated at 37 °C for 48 h. The inhibition zone size diameter was read by a vernier caliper. Furthermore, the negative control was 0.05% acetate buffer.

### 2.4. MIC of PLNC8 on H. pylori ZJC03

The MIC was calculated using the American Committee for Clinical Laboratory Standardization’s agar dilution method (NCCLS, 2000) [31]. Various concentrations of PLNC8 ranging from 0–160 μM were added to the CBA medium at 45–55 °C, and poured into plates with an inner diameter of 90 mm. A micro sampler was used to absorb 100 μL of diluted 1 × 10^8^ CFU/mL *H. pylori* ZJC03 suspension after cooling and solidification. The coating stick was applied evenly to the medium’s surface, and the Petri dishes were placed in a microaerobic bag and placed in a microaerobic environment at 37 °C for 48 h. The growth of *H. pylori* ZJC03 was observed and the lowest concentration of PLNC8 without bacterial growth in the medium was used as the MIC value [32]. Gently wash the bacteria, measure the absorbance at 550 nm, calculate the inhibition rate, and repeat all experiments three times. In addition, the negative control was 0.05% acetate buffer, and *H. pylori* ZJC03 Petri dishes were blank control.

### 2.5. Urease Activity Assay

Urease activity was detected by the phenol red discoloration method and used as an index to measure the viability of *H. pylori* ZJC03 [33,34,35]. *H. pylori* ZJC03 cells (OD_550_ = 0.50 ± 0.05) and different concentrations of PLNC8 (*v*/*v* = 1:1, 50 μL) in a Brucella Broth (BB) were mixed and put in 96-well plates. The mixture was co-cultured for 0, 2, 4, 6, and 8 h. After that, 100 μL of the combination samples were added to 100 μL of urea medium (0.1% yeast powder, 9.1% KH_2_PO_4_, 9.5% Na_2_HPO_4_, 20% urea, and 0.01% phenol red in ultra-pure water, pH = 6.5) in a new 96-well microtiter plate, and incubated under microaerobic conditions at 37 °C for 48 h. The value at OD_550_ was measured after 48 h.

### 2.6. Scanning Electron Microscopy

*H. pylori* ZJC03 was incubated at 37 °C under a micro-oxygen environment for 48 h and washed with BB broth (Hopebio, Qingdao, China), which was supplemented with 10% Calf Serum (yuanye Bio-Technology, Shanghai, China) and *H. pylori* selective supplement (Hopebio, Qingdao, China) to support the bacterial suspension for standby. *H. pylori* ZJC03 cells were treated with 1.0 × MIC of PLNC8 at 37 °C for 4 h under the microaerobic conditions. The untreated indicator *H. pylori* ZJC03 was used as a control. After being washed with PBS buffer (0.1 M, pH = 7.4) (Sangon Biotech, Shanghai, China), *H. pylori* ZJC03 was prepared by adding 2 mL of 2.5% glutaraldehyde (Sinopharm Group Chemical Reagent, Shanghai, China) solution to a centrifuge tube, then fixation was performed overnight at 4 °C. *H. pylori* ZJC03 was diluted with PBS buffer for 15 min to dehydrate and added with 1% osmic acid (SPI Supplies, USA) for 1–2 h. Attenuated with PBS buffer three times. After that the sample was dehydrated with 30%, 50%, 70%, 80%, 90%, 95% and 100% ethanol (Sinopharm Group Chemical Reagent, China) for 15 min. After being fully desiccated in the critical point dryer (Hitachi HCP-2), the dehydrated sample was coated with gold-palladium in Hitachi Model E-1010 ion sputter for 4–5 min as the 10 nm gold film. The treated sample was observed in Hitachi Model SU-8010 SEM.

### 2.7. Transmission Electron Microscopy

The pretreatment method was the same as in Section 2.6. The sample was treated with ager for embedding and fixed with 1% osmic acid for 1 h; 30%, 50%, 70%, 80% ethanol, 90% and 95% acetone (Sinopharm Group Chemical Reagent, China) were successively used for dehydration for 15 min. The two separate dehydration sessions of 20 min each with 100% acetone were performed. After elution, the sample was processed. Put the specimen in a 1:1 (*v*/*v*) mixture of 100% acetone and Spurr (SPI Supplies, USA) resin mixture at room temperature for 1 h, transferred the specimen to a 1:3 (*v*/*v*) mixture at room temperature for 3 h and placed the sample in Spurr resin mixture overnight. After the permeation treatment, the sample was heated at 70 °C for more than 9 h to obtain the embedded sample. Sections were cut to 80 ± 10 mm in the Leica EM UC7, treated with dye lead citrate solution (Sinopharm Group Chemical Reagent, Shanghai, China), and the saturation of uranyl acetate (SPI Supplies, West Chester, PA, USA) solution for 5–10 min. Photograph with Hitachi H-7650.

### 2.8. Relative ATP Levels

*H. pylori* ZJC03 were treated with 0, 0.25, 0.5, 1.0, 2.0, and 3.0×MIC PLNC8 for 4 h. Aliquots of negative control (0.05% acetate buffer) and positive control (1% Triton X-100 solution) [36,37]. The sample was placed in 25 μL ATP test lysate and 100 μL ATP test reagent (Beyotime Biotechnology, Shanghai, China) and then mixed into tubes for 30 min. Afterward, washed the cells and used passive lysis buffer to lyse them. The lysate tubes were centrifuged at 12,000 rpm/min for 5 min at 4 °C, and the supernatant was collected. Added the supernatant to a new black 96-well plate. The fluorescence intensity was measured with the multimode reader.

### 2.9. Hydrogen Peroxide Sensitivity Assay

Based on the method of Harris et al. [38,39], some improvements are made. *H. pylori* ZJC03 was treated with or without 0.25, 0.5, 1.0, 2.0, or 3.0×MIC PLNC8 for 12 h. 500 μL *H. pylori* ZJC03 and PLNC8 culture samples were mixed with 1.5 mL of 3% hydrogen peroxide solution. 100 µL of the mixture was quickly removed every 3 min and mixed with 1% bovine liver catalase solution of the same volume (Sigma Chemical Co., Shanghai, China). Samples were diluted in serial ten-fold dilutions, and 100 µL dilute samples were spread over more than three CBA plates cultivation. Incubated the plates at 37 °C under the microaerobic conditions and determined the number of colony-forming units (CFU/mL) after 72 h.

### 2.10. Statistical Analysis

All the data are symbolized as the average ± standard deviation (SD). Using GraphPad Prism version 8.0 software (GraphPad Software Inc., San Diego, CA, USA) analyzed all data by t-tests (and nonparametric tests). Consequences are defined as statistically significant at * *p* < 0.05, ** *p* < 0.01, and *** *p* < 0.001.

## 3. Results and Discussion

### 3.1. Inhibitory Effect of PLNC8 on the Growth of H. pylori ZJC03

Measuring the bacteriostatic circle was the most intuitive method to detect whether the sample had the effect of inhibiting pathogenic bacteria [40]. Figure 1 shows the inhibition ability of various concentrations of PLNC8 against *H. pylori* ZJC03 and the diameter of its inhibition zone. The inhibition zones of high concentrations of PLNC8 were more obvious, indicating that PLNC8 had the potential ability to inhibit *H. pylori* ZJC03 growth. As shown in Figure 1B, an obvious transparent inhibition zone was observed at a concentration of 80 μM (7.06 ± 0.15 mm) but not at a concentration of 64 μM (6.00 ± 0.00 mm). It was observed in the inhibition zone that the inhibition concentration of PLNC8 on *H. pylori* ZJC03 was between 0–160 μM. In order to further verify its MIC, the method described in Section 2.4 was repeated three times and the inhibition rate of PLNC8 was determined. The results are shown in Figure 2. The MIC was defined as the lowest concentration at which the indicator bacterium showed an extremely small number of colonies or no growth [31,41]. Co-culture of PLNC8 with *H. pylori* ZJC03 showed that the MIC of PLNC8 was 80 μM. There was no detectable growth at 37 °C after 72 h.

### 3.2. Urease Activity of H. pylori ZJC03

In vitro, *H. pylori* generally cannot exist in environmental conditions below pH = 3.0. If a urea concentration similar to the gastric cavity environment is added, *H. pylori* is protected [42]. The analysis of the *H. pylori* gene sequence shows that it may be able to use several substrates as its nitrogen source, including urea, ammonia, and three amino acids (arginine, serine, and glutamate) Ammonia can be produced by the decomposition of urea by urease [43]. This makes the nitrogen source exist in the form of ammonium ions. It seems that *H. pylori* can also encode fatty amidase [44], which catalyzes amide decomposition and provides a nitrogen source for bacterial metabolism through ammonia production. Therefore, the urease produced by *H. pylori* allows it to survive in an acidic environment. *H. pylori* produced urease and was considered a major causative factor in gastritis and peptic ulcers (a small painful area in the human stomach). As shown in Figure 3, the treatment with different concentrations of PLNC8, showed that the urease activity of *H. pylori* was negatively correlated with PLNC8 concentration. When the concentration of PLNC8 increased, the urease activity of *H. pylori* ZJC03 was relatively decreased, from 98.58 ± 0.86% (0.05% acetate buffer) to 25.68 ± 1.36% (3.0×MIC), and the urease activity was stable after the *H. pylori* ZJC03 strain had been treated with PLNC8 for 4 h. It was speculated that PLNC8 inhibits the growth of *H. pylori* ZJC03 and that the measured urease activity was lower than that of the control group. Therefore, PLNC8 can inhibit the growth of *H. pylori* ZJC03.

### 3.3. In Vitro Mechanism of Action of PLNC8 on H. pylori ZJC03

#### 3.3.1. Microscopic Images of *H. pylori* ZJC03

SEM and TEM were used to examine the microscopic structure of *H. pylori* ZJC03 and observe whether the cell structure was damaged [28]. Based on the results of experiments 3.1 and 3.2, PLNC8 inhibited *H. pylori* ZJC03 growth. The changes in the microstructure after *H. pylori* ZJC03 was exposed to 80 μM PLNC8 for 4 h were evaluated by electron microscopy (SEM and TEM) (Figure 4). Untreated *H. pylori* ZJC03 cells had a complete screw shape (Figure 4A,B). After treatment with 80 μM PLNC8, the majority of *H. pylori* ZJC03 cells had broken or vanished structures, and some even collapsed into cell debris (Figure 4C,D). On evaluation by TEM, the untreated *H. pylori* ZJC03 cells had a complete cell wall and dense cytoplasm. Furthermore, distinct boundaries between the cell wall and cytoplasm were observed (Figure 4E,F). After *H. pylori* ZJC03 cells were treated with 80 μM PLNC8, the cell wall and cytomembrane were recessed, and a portion of the cytoplasmic contents was detached from the cytomembrane (Figure 4G,H). Based on the results in Figure 4, it was speculated that PLNC8 may destroy the cytomembrane structure of *H. pylori* ZJC03, leading to the leakage of cell contents and thereby achieving a bactericidal effect.

#### 3.3.2. The ATP Levels in *H. pylori* ZJC03

Bacteriocins induce membrane permeabilization of target bacteria, presumably through the formation of ion-selective pores leading to loss of the proton motive force, intracellular ATP depletion, leakage of intracellular substrates, and ultimately death [45]. ATP was the direct energy source of cells, providing energy for life activities [36]. Three proton-transporting ATPases have been confirmed in *H. pylori*: ATPase-439, ATPase-948, and ATPase-115 [46]. In addition, Hazell et al. [47] found fumarate reductase in *H. pylori*, suggesting that *H. pylori* can obtain ATP through anaerobic respiration, similar to other anaerobic bacteria and facultative anaerobic bacteria. Meanwhile, Tatusov et al. proposed that since the most common respiratory quinone is 6-methylquinone, anaerobic respiration is more common than aerobic respiration in *H. pylori* [48]. When the cell undergoes necrosis or apoptosis, the intracellular ATP concentration decreases to some extent, so the ATP level reflects the cell state to some extent [37]. As the results presented in Figure 5 clearly indicated, when PLNC8 was applied to *H. pylori* ZJC03, the ATP levels decreased, showing that PLNC8 inhibited *H. pylori* ZJC03 growth, and the 1.0×MIC PLNC8 treatment of *H. pylori* resulted in approximately 3.7 times lower *H. pylori* ZJC03 growth than no treatment. In conclusion, disrupting the ATP levels of *H. pylori* ZJC03 to reduce the energy of *H. pylori* ZJC03 growth may be one of the modes of action of PLNC8.

#### 3.3.3. Hydrogen Peroxide Sensitivity

It has been shown that *H. pylori* infection led to increased levels of reactive oxygen species (ROS), oxidative DNA damage, and activation of oncogenic signaling pathways and is involved in gastric carcinogenesis [49,50]. Catalase was found to be potentially important for the growth and survival of *H. pylori*. The susceptibility of *H. pylori* ZJC03 to hydrogen peroxide lethality was assessed with and without PLNC8 treatment. Figure 6 shows that the viability of *H. pylori* ZJC03 treated with 1.0, 2.0, and 3.0×MIC PLNC8 disappeared within 9 min, whereas the viability of *H. pylori* ZJC03 without treatment decreased slightly and remained steady throughout the entire 21-min incubation period. The control group decreased slightly. Based on these results, it is speculated that *H. pylori* ZJC03 needs a strict microaerobic environment to maintain its activity. During the experiment, although *H. pylori* ZJC03 cells were able to change shape as a self-protection mechanism, some bacteria still died.

## 4. Conclusions

In this study, we evaluated the inhibitory effect of PLNC8 on *H. pylori* ZJC03 and confirmed that PLNC8 has a certain inhibitory effect on *H. pylori* ZJC03. In addition, a preliminary investigation and analysis of the mode of action of PLNC8 inhibition on *H. pylori* ZJC03 was conducted, and the results showed that PLNC8 may act on the cell membrane of *H. pylori* ZJC03 and the enzymes involved in maintaining the structure of the cell membrane to achieve its killing effect.

*Lactobacillus* bacteriocins have antagonistic effects on *H. pylori* and may inhibit *H. pylori* infection via a variety of mechanisms, including activation of autolysis enzymes. PLNC8 is a dipeptide bacteriocin produced by *Lactiplantibacillus plantarum*, and since both peptides are amphipathic and have a net positive charge, they interact with negatively charged phospholipids in the cell membrane of the target bacteria. Since the mechanism of action of PLNC8 is different from existing antibiotics and it has low toxicity to the host, it has become an interesting candidate for research. The present study only verified that PLNC8 has an inhibitory effect on *H. pylori*, its mechanism was first explored, and the specific mechanism of action has not been fully elucidated.

To further explore the mechanism of action of PLNC8, cellular and in vivo experiments will be conducted to investigate its effects on gastric micro-ecology and immune characteristics related to gastritis. This will aid in the development of a theory for the regulatory mechanism of *H. pylori* inhibition by functional substances. Overall, PLNC8 is expected to be a viable alternative to *H. pylori* ZJC03 infection prevention and treatment.

## Figures and Tables

**Figure 1 foods-11-01235-f001:**
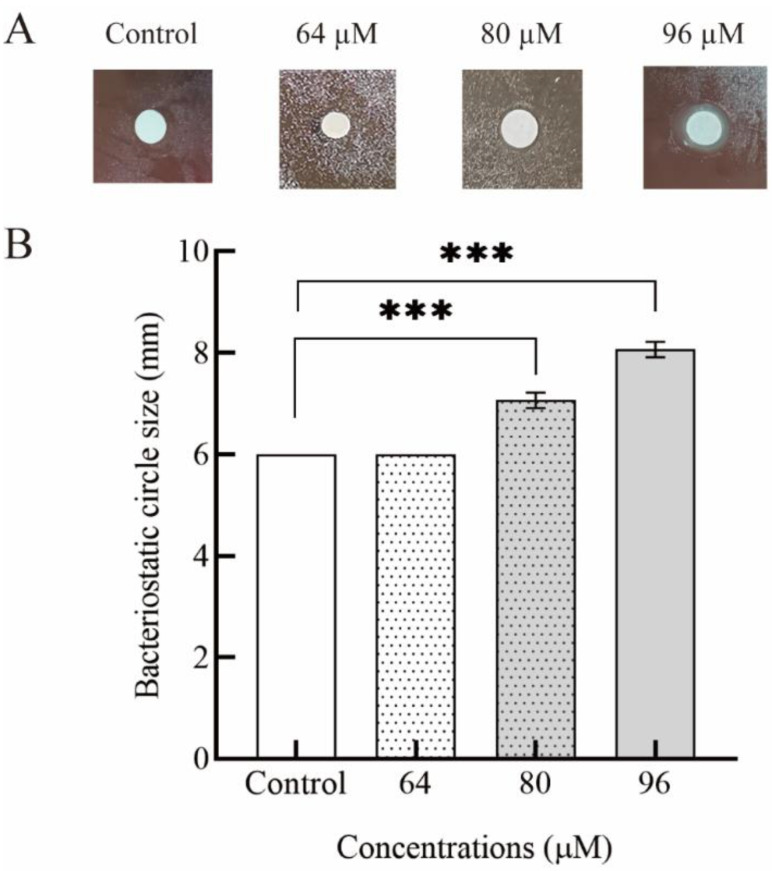
The diameter of the inhibition zone (DIZ) was determined by the filter paper diffusion method. The plates with *H. pylori* and PLNC8 were incubated at 37 °C in a microaerobic environment for 72 h. (**A**) Inhibitory effect of 0, 64, 80, and 96 μM PLNC8 on *H. pylori* ZJC03. (**B**) Inhibitory effect of different concentrations of PLNC8 on *H. pylori* ZJC03; Control: 0.05% acetic acid buffer; Data are shown as the mean ± SD; *n* = three groups parallel. *** *p* < 0.001; no statistical significance was shown.

**Figure 2 foods-11-01235-f002:**
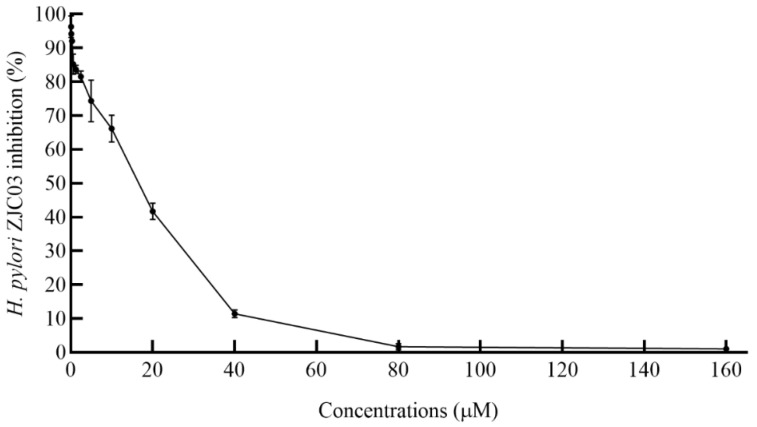
The MIC of PLNC8 against *H. pylori* ZJC03 was determined by the agar dilution method with a concentration range of 0–160 μM obtained by the twofold dilution method. The plates with the *H. pylori* co-cultured PLNC8 mixture were incubated at 37 °C in a microaerobic environment of 10% CO_2_, 10% O_2_, and 80% N_2_ for 72 h. The plate bacteria were gently washed down, and the bacterial suspension absorbance was measured at 550 nm to calculate the antibacterial rate.

**Figure 3 foods-11-01235-f003:**
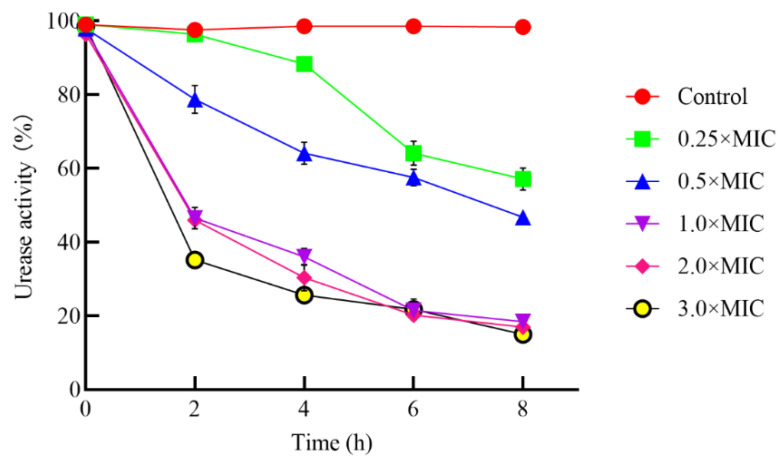
Urease activity of *H. pylori* ZJC03 co-cultured with or without different concentrations of PLNC8 for 0, 2, 4, 6, and 8 h; Control: 0.05% acetic acid buffer; Data are shown as the average ± SD.

**Figure 4 foods-11-01235-f004:**
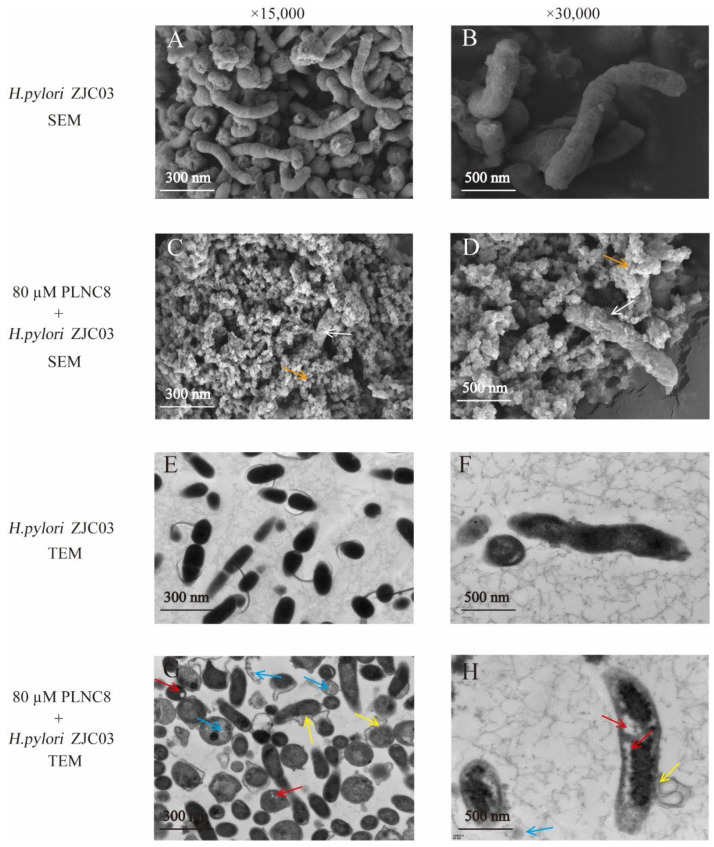
SEM and TEM were used to observe the microscopic structural changes in *H. pylori* ZJC03 treated with PLNC8 after 4 h at 37 °C. SEM (×15,000) observation is shown in pictures (**A**,**C**). SEM (×30,000) observation is shown in pictures (**B**,**D**). TEM (×15,000) observation is shown in pictures (**E**,**G**). TEM (×30,000) observation is shown in pictures (**F**,**H**). (**A**,**B**,**E**,**F**) *H. pylori* ZJC03 cells without treatment. (**C**,**D**,**G**,**H**) *H. pylori* ZJC03 cells treated with PLNC8. Broken cytoplasm (blue arrow), cell wall separation (yellow arrow), cell collapse (white arrow), cell lysis (orange), and cell perforation (red arrow).

**Figure 5 foods-11-01235-f005:**
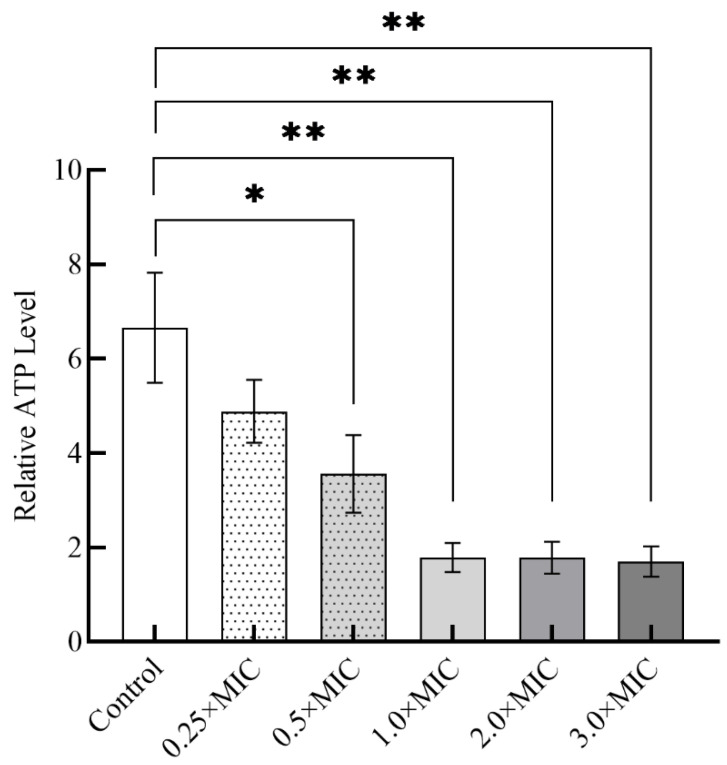
Effects on ATP levels in *H. pylori* ZJC03. ATP of *H. pylori* ZJC03 co-cultured with different concentrations of PLNC8 for 30 min; Control: 0.05% acetic acid buffer; Data are shown as the average ± SD; *n* = three groups parallel. * *p* < 0.05 and ** *p* < 0.01; there was no significant difference if it was not marked.

**Figure 6 foods-11-01235-f006:**
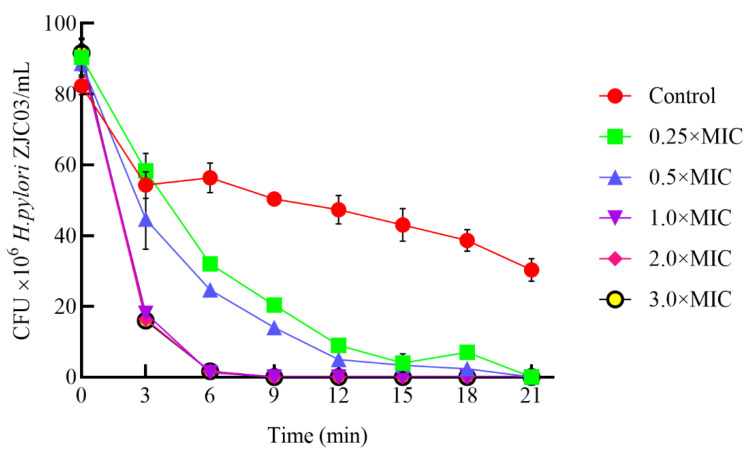
Hydrogen peroxide sensitivity of *H. pylori* ZJC03. Broth cultures of *H. pylori* ZJC03 treated with or without 0.25, 0.5, 1.0, 2.0, or 3.0×MIC PLNC8 were added to 100 mmoL/kg hydrogen peroxide solution and evaluated by CFU determination every 3 min for 21 min in total; Control: 0.05% acetic acid buffer.

## Data Availability

The datasets generated for this study are available on request to the corresponding author.

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
