# Peer review of "Antibacterial Effects of Bacteriocin PLNC8 against Helicobacter pylori and Its Potential Mechanism of Action"

_foods, 2022, doi:10.3390/foods11091235_

Round 1

Reviewer 1 Report

Dear authors, the manuscript has improved substantially. I still think that the images in Figure 1A should have better resolution. Please, % and degrees Celsius (C) must be separated from the number.

Reviewer 2 Report

The authors have included all comments in the revised manuscript.

Author Response

Thank you for your letter and the reviewers’ comments on our manuscript entitled " Antibacterial effects of bacteriocin PLNC8 against Helicobacter pylori and its potential mechanism of action" (foods-1695008). Those comments are very helpful for revising and improving our manuscript, and we have made corrections which we hope meet with approval. Revised portions are marked in red in the manuscript.

The point to point responds to the reviewer’s comment is listed as following:

Comment 1: The authors have included all comments in the revised manuscript.

Response: We really appreciate for the reviewer’s comment. The typos and grammatical errors through the manuscript were corrected as suggested.

This manuscript is a resubmission of an earlier submission. The following is a list of the peer review reports and author responses from that submission.

Round 1

Reviewer 1 Report

Dear Authors,

This phrase - Lines 238 -240 " In conclusion, disruption of the mitochondria of H. pylori ZJC03 to inhibit the growth of H. pylori ZJC03 may be one of the modes of action of PLNC8" must be rewritten correctly.

Is the use of PLNC8 in eukaryotic cells safe? 

Best regards,

Reviewer 2 Report

This study demonstrated the antimicrobial activity of the bacteriocin PLNC8 produced by Lactobacillus plantarum ZJ316 against Helicobacter pylori. To investigate the mechanism behind this activity, the authors examined the effect of PLNC on the H. pylori cell structure, ATP level, urease activity, and sensitivity to hydrogen peroxide. While the characterization of a bacteriocin against H. pylori could be of great interest for public health given the emergence of antibiotic-resistant pathogens, the current manuscript is overall poorly written and has revealed serious weaknesses in various aspects.

  1. Numerous grammatical errors, fragments, and unclear statements have greatly hampered readability and must be addressed to improve the manuscript.
  2. Experimental procedures are not described in sufficient detail and the data are not adequately presented in the Results and Discussion with some examples shown below.

          - Lines 206-208: Clarity these statements since they seem to suggest that the determination of the antibiotic activity of PLNC against H. pylori and electron microscopy analysis were already conducted in a prior study. If this is the case, the novelty of this study would be quite limited.

          - Lines 232-235: I am not sure whether these statements regarding mitochondria, necrosis, and apoptosis are relevant here since this paper pertains to prokaryotes, not eukaryotes. Also, in lines 238-240, clarify "disruption of the mitochondria of H. pylori ZJC03". Does this mean that this H. pylori strain has a mitochondrion?

          - Line 250: This PCR result does not guarantee that ureC gene is functional and responsible for urease production since genetic changes including premature stop codons and frameshifts can be present.

          - Lines 257-259: Unless urease activity is required for the growth of H. pylori, it sounds strange to conclude that the growth inhibition is due to the negative effect of PLNC on the urease activity of H. pylori.

          - Lines 269-271: This statement is not relevant here. Remove or revise it.

          - Lines 274 and 275: "the viability of H. pylori ZJC03 without ... period" does not match the data presented in Figure 6 since the cell number also decreased, albeit less rapidly, without the bacteriocin.

  1. References are often missing as exemplified in lines 37-39, 42-43, 206, and 246-247. Provide references as often as needed.
  2. Figure legends are not detailed enough, which could confuse readers. For instance, in Figure 3, two pictures were presented for the same sample analyzed with the same technique, for instance, Figures 3A and 3B for untreated H. pylori cells examined with SEM. Although they seem to differ in magnitude, it is not mentioned at all in the figure legend, which initially confused this reviewer greatly and will do for many readers.
  3. I suggest eliminating the statements that merely repeat the figure titles, which are often found in the Results and Discussion section.
  4. While the authors provide information on H. pylori strain ZJC03, they did not mention why this strain was chosen to assess the antimicrobial effect of PLNC8.

Reviewer 3 Report

Synthesis and preparation of PLNC8 α and PLNC8 β were not clearly described. The authors need to explain how PLNC8 was synthesized in brief.

For the microbiological analysis of PLNC8 by Zone of inhibition and determination of MIC, did the authors use any positive and negative control? It should be indicated in the materials and methods.

Figure 1 A shows that 96 μM concentration has a clear zone around the disc, and Figure 2 shows the MIC value is 96 μM. Why authors “speculated that 80 μM might be the minimum inhibition concentration”. Authors need to justify the antimicrobial results.

 Figure 3. Figure Caption is very confusing,  what sample is A, B, C, D,...

All pictures should be in the same magnification to compare the results control to the test sample.

“Broken cytoplasm (green arrow) is not shown in the pictures..

The SEM results and TEM results do not clearly show that there is actual damage of “cytomembrane (yellow arrow) and the channels of the membrane (red arrow)”. To know the damage of the cytomembrane authors should use some fluorescence techniques, which highlighted the damage of cytomembrane.

The conclusion is very superficial written, it needs to be revised.